# The relationship between trust and attitudes towards the COVID-19 digital contact-tracing app in the UK

Liz Dowthwaite[1]*, Hanne Gesine Wagner[1,2], Camilla May Babbage[3], Joel E. Fischer[2], Pepita Barnard[1], Elena Nichele[1], Elvira Perez Vallejos[4], Jeremie Clos[5], Virginia Portillo[1], Derek McAuley[1]

1 Horizon Digital Economy Research, University of Nottingham, Nottingham, Nottinghamshire, United Kingdom, 2 Mixed Reality Lab, School of Computer Science, University of Nottingham, Nottingham, United Kingdom, 3 NIHR MindTech MedTech Co-operative, School of Medicine, University of Nottingham, Nottinghamshire, United Kingdom, 4 Division of Psychiatry and Applied Psychology, Nottingham University, Nottingham, United Kingdom, 5 School of Computer Science, University of Nottingham, Nottingham, Nottinghamshire, United Kingdom

* liz.dowthwaite@nottingham.ac.uk

**Data Availability Statement:** All questionnaire data are available from the University of Nottingham Research Data Management Repository (DOI: 10.

## Abstract

During the COVID-19 pandemic, digital contact-tracing has been employed in many countries to monitor and manage the spread of the disease. However, to be effective such a system must be adopted by a substantial proportion of the population; therefore, public trust plays a key role. This paper examines the NHS COVID-19 smartphone app, the digital contact-tracing solution in the UK. A series of interviews were carried out prior to the app's release (n = 12) and a large scale survey examining attitudes towards the app (n = 1,001) was carried out after release. Extending previous work reporting high level attitudes towards the app, this paper shows that prevailing negative attitudes prior to release persisted, and affected the subsequent use of the app. They also show significant relationships between trust, app features, and the wider social and societal context. There is lower trust amongst non-users of the app and trust correlates to many other aspects of the app, a lack of trust could hinder adoption and effectiveness of digital contact-tracing. The design of technology requiring wide uptake, e.g., for public health, should embed considerations of the complexities of trust and the context in which the technology will be used.

## Introduction

The World Health Organization (WHO) recommends that contact-tracing should trace and quarantine 80% of close contacts within 3 days to be deemed successful to diminish the spread of COVID-19 [1]. Therefore, public participation in and adoption of contact-tracing measures plays a vital part in fighting the pandemic [2] In the UK, the official contact-tracing smartphone app was launched by the NHS (National Health Service, the publicly funded healthcare system in England) in September 2020 to much anticipation after delays and controversies surrounding its development in the previous months [3, 4]. The "NHS COVID-19" app employs

17639/nott.7223). Excerpts of the transcripts are provided in the manuscript.

**Funding:** This research was supported by the Engineering and Physical Sciences Research Council (https://www.ukri.org/councils/epsrc/), grants EP/V00784X/1, EP/M02315X/1, and EP/T022493/1. EPV and CMB also acknowledge the resources of the National Institute for Health Research, Nottingham Biomedical Research Centre. The funders had no role in study design, data collection and analysis, decision to publish, or preparation of the manuscript.

**Competing interests:** The authors have declared that no competing interests exist.

Bluetooth proximity technology to identify close contacts with those who tested positive for the virus, and automatically alerts people that they should self-isolate at home for a period of up to 10 days if they were a close contact [5]. Thus, while adopting the contact-tracing app may be important to alleviate the spread of the virus, it may also cause severe disruption to people's lives such as by impacting livelihoods, care responsibilities, and mental health—more than 600,000 people were alerted to self-isolate in a single week in July 2021 alone, causing severe service disruptions and workforce shortages [6].

This paper is an exploratory look at how the public (in the UK) feel about contact tracing, and the Test and Trace App in particular. so our aims for the paper were twofold: firstly, to explore how public feeling about the potential introduction of contact tracing in response to the pandemic relates to how they actually felt once such a solution was introduced, and secondly, to add greater depth to our understanding of how exactly trust relates to the use and attitudes towards the NHS COVID-19 smartphone app, the digital contact-tracing solution in the UK, once it was released. This paper provides a qualitative contextualization by means of a thematic analysis of previously unreported interview data, describing the prevailing attitudes in the UK around the time of the pandemic, and how they felt about the introduction of contact tracing. As well as understanding how people feel about using the app, it is important to understand how people felt about contact tracing prior to the app being released. This will give insight for future public health interventions to understand and avoid the potential issues, and promote uptake on release. The paper also expands upon the findings of prior work [7], which showed that there are issues surrounding trust and understanding that hindered adoption and therefore the effectiveness of the NHS COVID-19 smartphone app. Whilst that paper focused on the differences in attitudes between potentially vulnerable (participants who were over 65 years old or Black, Asian and Minority Ethnic) and users who were not at higher risk to contract COVID-19, this paper examines more closely the relationships between trust and the drivers and barriers to the adoption of digital contact-tracing, through new statistical analyses. By using an overall score for Trust in the App, we are able understand the relationships between trust and motivations for download, trust in others, and specific aspects of the app itself which were previously unreported. These additional analyses reveal that trust has significant effects on both use and non-use on several levels; key findings include:

1. The strongest drivers for downloading the app, including wishing to help the NHS, reduce the spread of the virus, and protect oneself and others, show moderate correlations with levels of trust.

2. The more participants trust the institutions involved in contact-tracing, including the UK government, local government, big tech companies, private contractors, the NHS, and large hospitality venues, the greater their trust in the app.

3. Higher levels of trust in the app were also related to higher importance placed on various features of the app by respondents, for example that the app provided explanations for information given to them, that they could verify that notifications were authentic, and that they could speak to a person about any advice from the app.

4. Higher levels of trust were related to positive attitudes towards the app, especially that it was reliable, useful to them and wider society, and easy to use.

5. A lack of trust in those who built the app is among the most common reasons for not downloading it, along with not wanting to be tracked, not thinking it would be effective, and not wanting to take part in contact-tracing in that way.

In our discussion we highlight the ways in which trust is a key factor in adoption and non-adoption of digital contact-tracing, ranging from views on technical aspects of the app itself to broader social and societal issues concerning family and loved ones and trust in institutions at the centre of the pandemic response (e.g. the NHS, the UK government). Our findings emphasise the importance of considering trust deeply in technology adoption research, and point towards participatory approaches engaging communities as potential starting points through which HCI research could contribute to a more inclusive design process of digital contact-tracing.

Herein we briefly review recent research on digital contact-tracing, particularly work that examines technology acceptance, adoption, and trust. We also briefly provide an overview of the pandemic in the UK as context.

### Digital contact-tracing

Digital contact-tracing solutions have been widely used to monitor and manage the spread of disease during the COVID-19 pandemic, but for it to be effective in reducing the impact of the pandemic, it must be adopted by a substantial proportion of the population [1, 8, 9]. In the UK, Ireland, the EU, and the US, public acceptance of app-based contact-tracing has been shown to be high [10–14], but it is vital to understand specific drivers and barriers to the use of digital contact-tracing which may be drawn upon to improve their design and increase uptake as a result [15]; this research sets out to do this in the UK context.

Studies of hypothetical digital contact-tracing apps suggested the main drivers for adoption in the UK include stopping the pandemic and protecting family, friends, and their community; potential barriers include increasing anxiety about the pandemic, as well as fear of hacking and increased surveillance after the pandemic [10, 13]. A study of hypothetical acceptance in Wales also found controlling the disease and supporting others to be the strongest reasons for app use, whilst mistrust in the government, concerns about data, and not thinking it would be an effective intervention were reasons against use [16]. Several studies used the Technology Acceptance Model (TAM2) [17] to explore acceptance of hypothetical tracing apps, for example in Switzerland perceived effectiveness and user-overall experience of contact-tracing apps depended on it being embedded in the health system [18]. In Germany, comparing a contact-tracing app and a data donation app found that motivations for use and perceived utility were higher for the contact-tracing app [19]. Another study suggested that concerns about health might override other concerns such as privacy [20]. However, in Australia nearly 28% of participants in a study of 1500 refused to download the COVIDSafe app, citing privacy and technical concerns, the belief it was unnecessary due to social distancing, and distrust in the government [15]. Participants in a UK study who did not download the contact-tracing app were also less likely to report that they understood how the app worked or that it was useful [7].

Acceptance of contact-tracing apps may also be influenced by the involvement or non-involvement of humans; a US study found that digital contact-tracing was preferred for preserving privacy, convenience, and accuracy, but human-based solutions can provide emotional assurance and advice [21]; others have also suggested that hybrid solutions are the best for acceptance [22].

However, studies have found differences between motivations related to intention to use, and those related to actual use [23] so it is important to study the adoption of live contact-tracing solutions. A study of the live app in the UK found similar motivations as for intended use, showing that the main drivers for download were wanting to help the NHS, protect themselves and others, and to reduce the spread of the virus; barriers to download were concern about

being tracked, a lack of trust in the people involved in the creation of the app, and not thinking the app would be effective [7]. In Ireland, the most common reasons for downloading the app were also linked to helping family and friends and a sense of responsibility to the wider community; barriers were related to trust, privacy and security, and fear of surveillance [14]. In the US, intention and actual use showed similar motives, including perceptions of the risk to health and surrounding health information, and perceived usefulness of the app [24].

**The UK context.** The two studies discussed in this paper consider attitudes towards digital contact-tracing at different times in the UK during the pandemic, as illustrated in Fig 1. Initial lockdown measures came into force in March 2020 before being relieved in June/July with some areas being subject to local lockdowns. At the end of August 2020, the month in which the first study was carried out, approximately 339,000 people in the UK had tested positive for COVID-19. The contact-tracing app had not been released yet, and all contact-tracing was done by human contact tracers as part of the overall Test and Trace system. The UK public were asked to 'check-in' to venues such as pubs and restaurants, which were open again over the summer into autumn, so that they could be contacted if positive cases were reported. Additionally, people who tested positive for COVID-19 were asked to give details of their recent contacts and where they had been; human contact tracers then contacted as many of those contacts as possible to warn them that they should isolate or get tested.

After a series of controversies surrounding its' development, the UK government released the NHS COVID-19 app on the 24th September 2020 [3]. The app is fully automated and decentralized, and uses self-reporting of symptoms and test results, Bluetooth proximity triggers, and venue check-ins using QR codes. By the time data collection was completed for the second study (21st December 2020), the number of positive tests had risen to approximately 2.18 million, or around 3.2% of the UK population [25]. The app has been shown to be effective in reducing the number of positive cases of COVID-19 in the UK, reducing the second wave by an estimated quarter [26]. As of February 2021, the app

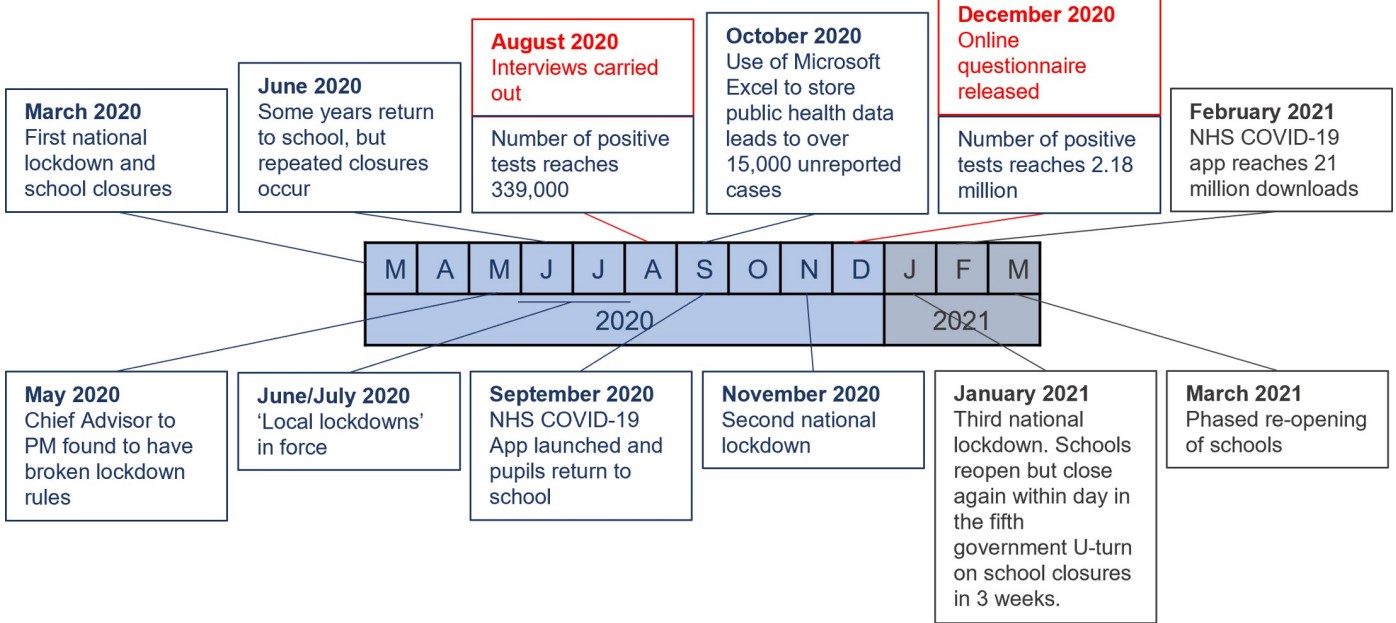

**Fig 1. A year of COVID-19 restrictions in the UK.** A summary of events in the UK surrounding COVID-19 between March 2020 and March 2021, with the timing of the Interview and Questionnaire studies included.

had been downloaded more than 21 million times, suggesting a 56% uptake among smartphone owners over 16 years old in the UK, and 1.7 million isolation alerts had been sent out [27, 28]; however in the period leading up to the second study reported in this paper, only 28% of people had actively used the app.

## Trust and acceptance of digital contact-tracing

Trust is an important factor in people's lives, being a consistent predictor of subjective wellbeing [29], and associated with overall life satisfaction [30, 31]. It is also extremely complex, being highly context specific and multidimensional [32]. Trust between humans may be considered a mental state that is felt by a person (or not) with regards to another entity [33]; trust in technology such as information systems may be considered similarly as it is often based on trust in the humans and organisations that control them, rather than the system itself [34]. The motivation to trust is also related to the willingness to continue to trust or to restore trust [35], implying that when something is important to someone, that person is more willing to place trust in that situation, thing, or person.

Trust may significantly impact the adoption of contact-tracing apps [14, 36]. Higher levels of trust in the government and in health authorities may lead to increased uptake of such an app [18]; distrust in the government was a factor in decisions not to download the Australian contact-tracing app COVIDSafe [15]. In Germany, both general trust in official app providers and social trust played an important role in perceptions of digital contact-tracing [19], and across five countries (France, Germany, Italy, the UK and the USA) lack of trust was found to be one of the main barriers to adoption of a hypothetical contact-tracing app [10]. In the UK, it was found that people who chose not to download the NHS COVID-19 app reported significantly lower trust in the app, including in the use of data and that the app would work as it was supposed to, as well as lower trust in other people to download the app or to self-isolate if they were told to by the app; they were also significantly less trusting of stakeholders [7]. Findings about the importance of embedding digital contact-tracing into the health system [18] suggest that trust and confidence in the NHS might influence attitudes towards the app and its usage in the UK. Distrust has also been found to be a factor in non-adoption of digital contact tracing [37].

## Method 1: Interviews

Ethical approval was granted by the Computer Science Research Ethics Committee at the University of Nottingham; no minors were involved in the study. Participants were adults, who were provided with information and privacy notices and filled out an online written consent form (including confirming that they were over 18 years of age) prior to taking part in the interview. At the start of the interview, verbal consent was taken by the interviewer and recorded as part of the interview, in addition to the online consent form., this was approved by the ethics committee. Interviews were carried out in the summer of 2020, prior to the release of the NHS COVID-19 app.

## Participants

Twelve interviews were carried out with members of the public residing in the UK and over 18 years of age. Participants were recruited via email and advertising on social media and were incentivized with a 15GBP shopping voucher. Individual details of participants are not provided to safeguard their privacy; participants were 6 males and 6 females, 10 had at least an undergraduate degree, and 10 were white. Eight participants resided in England, 2 in Wales and 2 in Scotland. When quoted in the text, participants are identified by a single letter from A-L.

### Procedure

Interviews were carried out online via Microsoft Teams in August 2020, between "lockdown 1" being eased in June and new restrictions being put in place at the start of September. Interviews were carried out by an experienced socio-technical researcher. The interviews followed an open-ended, semi-structured format, designed to allow participants to say as much or as little as they wished about a series of topics surrounding Test and Trace in the UK. The NHS App had not yet been released, so the interviews began by broadly discussing the current Test and Trace system in the UK, which involved 'manual' contact-tracing carried out by human tracers. Then participants were asked what they had heard and understood about a potential digital contact-tracing app. This included any media stories they had heard, their understanding of how it would work, and whether and why they would use and trust such a system. Finally, participants were asked about the broader ecosystem of Test and Trace, including how it had been done in other countries, and how the NHS and the government had dealt with the pandemic. The interviews lasted between 25 minutes and 72 minutes, with an average of 46 minutes. A pilot interview with a member of the public was carried out prior to the main interviews to check for understanding of questions, timing, and context.

### Analysis

All interviews were video-recorded and automatically transcribed using Microsoft Streams, and then corrected by three of the authors. Responses were thematically analysed using Microsoft Excel, following the principles of Braun and Clarke [38, 39]. Applying a reflexive thematic approach to the research, both researchers kept a reflective journal throughout the analysis and were open with one another about potential influences, such as culture, feelings and experiences that might alter the interpretation of the data. This was especially important given the pandemic being a personal and unavoidable experience for all researchers.

Two researchers worked collaboratively to complete the qualitative analysis, coming from different academic backgrounds and both with previous training and widespread experience of thematic analysis. Themes and sub-themes were generated from the data, and in doing so commonalities were noted between the survey and qualitative data. To understand the relationship between trust and attitudes towards the COVID tracing app, a mixed methods approach was applied to this paper with supporting quotes taken from the qualitative analysis and inspiration from the themes that had been developed, specifically related to trust in technology. The themes and subthemes will be reported elsewhere.

Regular data sessions were held with other members of the team to ensure clarity and consistency of the coding and the themes, and how this related to the survey data. The quotes in this paper are taken from the pertinent themes identified by the analysis.

## Method 2: Questionnaire

Ethical approval was granted by the Computer Science Research Ethics Committee of the University of Nottingham. Participants were provided with information and privacy notices and gave informed written consent to take part; no participants were under the age of 16 and for those under 18 no additional parental consent was required, as agreed by the ethics board. A large-scale online questionnaire was carried out in December 2020, once the app had been available for a period of approximately 3 months. Whilst the interviews aimed to understand factors in peoples' decisions to make use (or not) of a contact tracing app, the questionnaire extended on this by examining attitudes towards the contact tracing app after release.

## Participants

Recruitment for study 2 was carried out by Ipsos MORI, a market research agency, via email to a randomly selected pool of online panel members meeting relevant criteria to gain a nationally representative sample based on age, gender, and region; there was also a 10–15% quota for Black and Minority Ethnic (BAME) respondents. A total of 2,575 invitations were sent out and 1,001 participant aged 16–75 years old took part. Participants were incentivized for participation with monetary compensation paid into their panel account. No personally identifiable information on the participants was taken but they were asked several demographic questions, as summarized in Table 1. Very few (4.0%, n = 40) participants reported that they had tested positive for COVID-19 or had been asked to self-isolate (8.5%, n = 85). Some stated that at least one member of their household (family member 6.4% (n = 64); non-family member 8.7% (n = 87) or someone close to them outside their household (family member 14.2% (n = 142); non-family member 17.1% (n = 171)) had tested positive. More than half of the participants (55.6%, n = 556) had experienced none of these things.

## Procedure

The questionnaire was carried out online between 11th and 21st December 2020, when the UK was subject to a regional tier system between "lockdown 2" and "lockdown 3". Questionnaire development was carried out in response to the themes surrounding trust generated by the interviews, in combination with elements of the Technology Acceptance Model (TAM2) [17]. Members of the author team developed a list of pertinent questions which were then tested and refined involving experts in questionnaire development from Ipsos MORI. Piloting of the

**Table 1. Summary characteristics of participants in the online questionnaire, n = 1,001.**

|  | Summary categories | Frequency | % |
|---|---|---|---|
| **Age** | 16–24 | 152 | 15.2 |
|  | 25–34 | 189 | 18.9 |
|  | 35–44 | 180 | 18.0 |
|  | 45–54 | 190 | 19.0 |
|  | 55–64 | 163 | 16.3 |
|  | 65–75 | 127 | 12.7 |
| **Gender** | Male | 501 | 50.0 |
|  | Female | 500 | 50.0 |
|  | Other | 0 | 0.0 |
| **Employment status** | Employed | 666 | 66.6 |
|  | Unemployed, including homemaker | 175 | 17.5 |
|  | Retired | 117 | 11.7 |
|  | Student | 43 | 4.3 |
| **Education** | Up to GCSE | 307 | 30.7 |
|  | Post-GCSE/A-level equivalent | 308 | 30.8 |
|  | Undergraduate and above | 386 | 38.6 |
| **Ethnicity** | White | 875 | 87.5 |
|  | Black, Asian, and Minority Ethnic | 115 | 11.5 |
|  | Not stated | 10 | 1.0 |
| **Country of Residence** | England | 847 | 84.7 |
|  | Wales | 48 | 4.8 |
|  | Scotland | 85 | 8.5 |
|  | Northern Ireland | 21 | 2.1 |

questionnaire and data collection was carried out by Ipsos MORI, who reviewed the data at n = 61 to check that collection was taking place correctly, and to check for understanding and anomalies. The data was reviewed again at n = 213 to ensure data quality.

All questions were closed-ended, either multiple choice or rated on Likert or Likert-like scales from 1 to 5 ("strongly disagree" to "strongly agree", or "not at all" to "entirely" as relevant); participants were routed to appropriate questions based on previous answers (Fig 2). The first part of the survey asked participants to indicate what knowledge and experiences they had of COVID-19 and the NHS contact-tracing app, for example if they had been asked to self-isolate and whether they had downloaded the app. The following section collected reasons for downloading and experiences of using the app amongst those who had downloaded it. Participants were then asked about app functionality and the technology involved, including whether the app was useful, easy to use, or beneficial, understanding of how it worked, and the importance of features such as opting in and out of contact-tracing. The following series of questions asked about levels of trust in distinct aspects of the app including responsibility, security, reliability, functionality, and data use. Overall trust in the NHS COVID-19 app was measured using the average score from an 8-item scale (Table 2), Cronbach's alpha = 0.91. Finally, participants were asked to rate their trust in a series of stakeholders including the government, the NHS, and small and large hospitality venues.

## Analysis

Responses were analysed using IBM SPSS statistics 26 and Microsoft Excel. Summary statistics (mean, standard error, standard deviation) or frequencies were extracted for all questions.

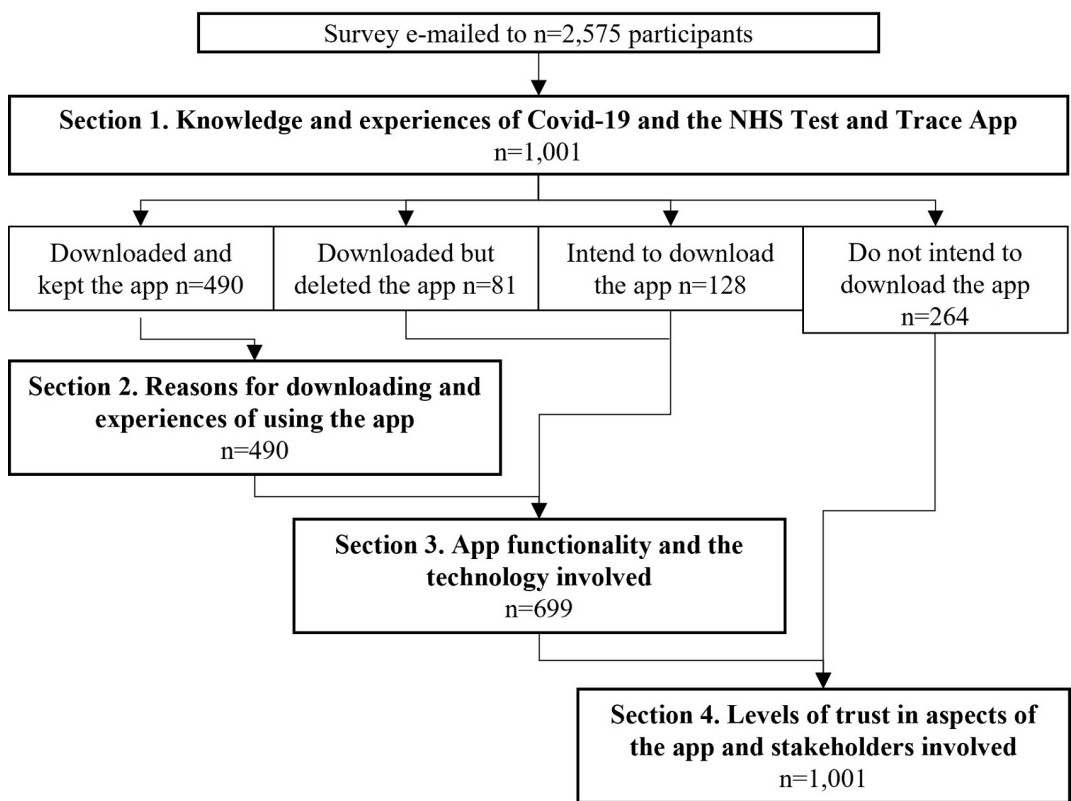

**Fig 2. Survey flow and branching of participants.** Specific questions/statements are reported in the text.

**Table 2. Trust in the NHS COVID-19 app.**

|  | Mean (SD) | Median (IQR) |
|---|---|---|
| **Overall trust in the NHS COVID-19 App** | 3.50 (0.88) | 3.63 (1) |
| **I trust that the data collected by the app is used responsibly** | 3.49 (1.17) | 4 (1) |
| **I trust that the data collected by the app is stored securely** | 3.48 (1.16) | 4 (1) |
| **I feel that the app is reliable** | 3.37 (1.16) | 4 (1) |
| **I trust that the app will do what it is supposed to do** | 3.49 (1.17) | 4 (1) |
| **I think the NHS COVID-19 app is basically trustworthy** | 3.56 (1.12) | 4 (1) |
| **I think that most other people will download the app** | 3.26 (1.11) | 3 (2) |
| **I trust that most other people will self-isolate if told to do so by the app** | 3.29 (1.15) | 3 (2) |
| **I trust that my data will be deleted when the app says it will** | 3.42 (1.19) | 4 (1) |

1 = Strongly disagree, 2 = Somewhat disagree, 3 = Neither agree nor disagree, 4 = Somewhat agree, 5 = Strongly agree. "Overall trust in the NHS COVID-19 App" is a composite made up of the average of the following 8 items, Cronbach's alpha = 0.91.

Confidence intervals for proportions are given at the 95% level. Missing data was reported as 'no response' and included in frequency calculations but excluded from inferential statistical analysis and the calculation of means. All inferential statistical analysis was carried out with a statistical significance threshold of $p < 0.05$. Most questions were significantly non-normal as shown by skewness and kurtosis, so non-parametric tests are appropriate. Kruskall-Wallis tests are used to examine trust in the app among different groups of users and Friedman's test is used to compare trust in institutions. A related-samples Friedman's two-way ANOVA is used to compare trust in different institutions. Spearman's rho correlations are carried out between trust in the app and responses related to technology aspects, reasons for downloading the app, and trust in institutions related to Test and Trace. A correlation above 0.2 is considered weak, above 0.4 considered moderate, and above 0.7 is considered strong; below 0.2 correlations are considered to indicate no relationship even if significant.

## Results

This section begins with an overview of the prevalent attitudes towards COVID-19 and contact-tracing in the UK, and how these related to trust, based on interview data, prior to the release of the app. This helps to provide the context of the ecosystem into which the app was released. This is followed by a summary of how questionnaire respondents felt about the NHS COVID-19 app in terms of trust after its release. The following sections lay out reasons for and against participation in contact-tracing relating to trust, how trust in the app relates to institutions involved in its deployment, and how trust in the app relates to its technological features. As the interviews are responses to a hypothetical app whilst the responses to the questionnaire are about actual app use, the results are organized with the qualitative data at the start, to consider how people felt (or expected to feel) about the app being released, followed by the questionnaire data to highlight how experiences of the app either back up or contradict initial opinions.

Just under half of the questionnaire participants (49.0%, n = 490) had downloaded the NHS COVID-19 mobile app and still had it on their phone. A further 12.8% (n = 128) had not yet downloaded it but intended to, 26.4% (n = 264) did not intend to download it, and 8.1% (n = 81) had downloaded it but since deleted it; 3.8% (n = 38) had not heard of the app. Most (92%, n = 451) had at least opened the app and had a look around, 66.7% (n = 327) had used it for venue check-in, 58.4% (n = 58.4%) had used the 'check symptoms' feature, and 71.2%

(n = 286) always had contact tracing (via Bluetooth) switched (a further 20.4%, n = 100, had it switched on some of the time).

## The Test and Trace ecosystem in the UK

Making any contact-tracing system (human, digital, or hybrid) work effectively is dependent on users' cooperation and behaviour, and this cooperation is dependent on a certain level of trust in those responsible for creating it. Interview participants' comments indicate that this was often a source of conflict for them. Reflecting on their experiences with two of the main stakeholders in the UK Test and Trace system, the NHS and the UK government, many participants expressed their sympathies for the health sector given the circumstances. *"I think they've [the NHS] done as best as they can with the resources they've got"* (Participant F). However, despite the NHS being highly dependent on the government, participants highlighted a difference in attitudes towards the two institutions, with the NHS often perceived in a better light. *"I think there was a lot of, in terms of public perceptions, there was a lot of government versus NHS kind of vibe at one point. [. . .] there was a lot of ill feeling about how carers were treated in terms of pay rates"* (Participant G).

The UK has several devolved areas, and differences in attitudes and trust towards the (local) government were also noted between different geographical locations. Attitudes toward the Scottish government for example were more favourable and trusting, in part due to different government attitudes towards local situations and expertise, *"The impression I get is that the Scottish government is more willing to rely on local knowledge expertise, whereas the British government wants to centralise things. Then it's more reluctant to give decision-making possibilities out"* (Participant B). Welsh participants also preferred for their own government to make the decisions, but not down to the local council level, who were seen as underequipped, *"I probably would prefer to come from Welsh government rather than coming from the local council 'cause I don't like the idea of there being different levels of quality and the other problem as well of course being how the Council was"* (Participant K). These geographic differences were strengthened with the introduction of different restriction levels between boroughs, *"I think the fact that it works differently in different places is a bit confusing to us normal people"* (Participant D).

The issue of trust and the willingness to download an (at the time hypothetical) contact-tracing app was discussed in the interviews. Generally, all interview participants were willing to download an app, saying that *"anything that helps protect us I'd say is a good thing"* and *"it's worth trying"* for this reason alone (Participant D). However, some participants voiced their doubts about data protection and what would happen to the collected data in the future. These ranged from anecdotal stories that (non-digital) tracing information had been used to contact individuals for non-COVID-related reasons (Participant B) to not trusting the government's (future) use of the data: *"I have concerns about how that data was used by the government. And whether or not they tie that data to testing data and then assume immunity or otherwise"* (Participant J). This was contrasted with others who pointed out that sharing data is *"all part of our daily life anyway and I think that lots of people give over so much information anyway"* (Participant E) so that using the contact-tracing app would not make a big difference. *"Anyone can get my name and my address and my telephone number from anywhere really, so I'm not sort of thinking oh you're gonna do something terrible to me if you get hold of my data if they should have it"* (Participant C).

Rather than privacy concerns, the greatest perceived threats were in the efficacy of contact-tracing systems were often related to the other people who would also need to use it. On the macro level, participants were aware of the dichotomy between a huge number of people needing to use the system to make it work versus the real potential that not all people would be able

to make use of it. *"One of my concerns with an app is [. . .] if you don't engage with it, it's kind of a useless thing. [. . .] I mean this is a really interesting generational question. [. . .] I suspect the coronavirus is more of an issue for elderly people than for younger people who would be much more comfortable with apps and with mobiles"* (Participant A). There was also seen to be a divide among different socioeconomic groups, for example those that cannot afford the technology (smart phones) needed to take part, or about dissatisfaction among certain parts of society: *"All through their lives they had to work hard and there it's always, but they had to fight against the system to get any help when they needed it and they haven't had the help when they've needed it. And then I think those people will say, well, Sod you"* (Participant H).

On a micro level, there was also distrust in others to do the right thing rather than what is easy and most beneficial to the individual, *"I think we're very selfish as a nation."* (Participant H). It was also felt that the lack of overall enforcement of the rules, led to a further tendency to act selfishly: *"Yes, it is a joint responsibility from everyone and it's just trying to get that through and trying to get enforced in some sort of way. It's just not been there unfortunately"* (Participant F).

## Overall trust in the NHS COVID-19 app

Among questionnaire respondents, overall trust in the NHS COVID-19 app is moderate (Fig 3, mean = 3.50, SD = 0.878). An Independent-Samples Kruskal-Wallis test shows a significant difference between trust in the app among the four groups of participants ($X^2(3) = 283.17$,

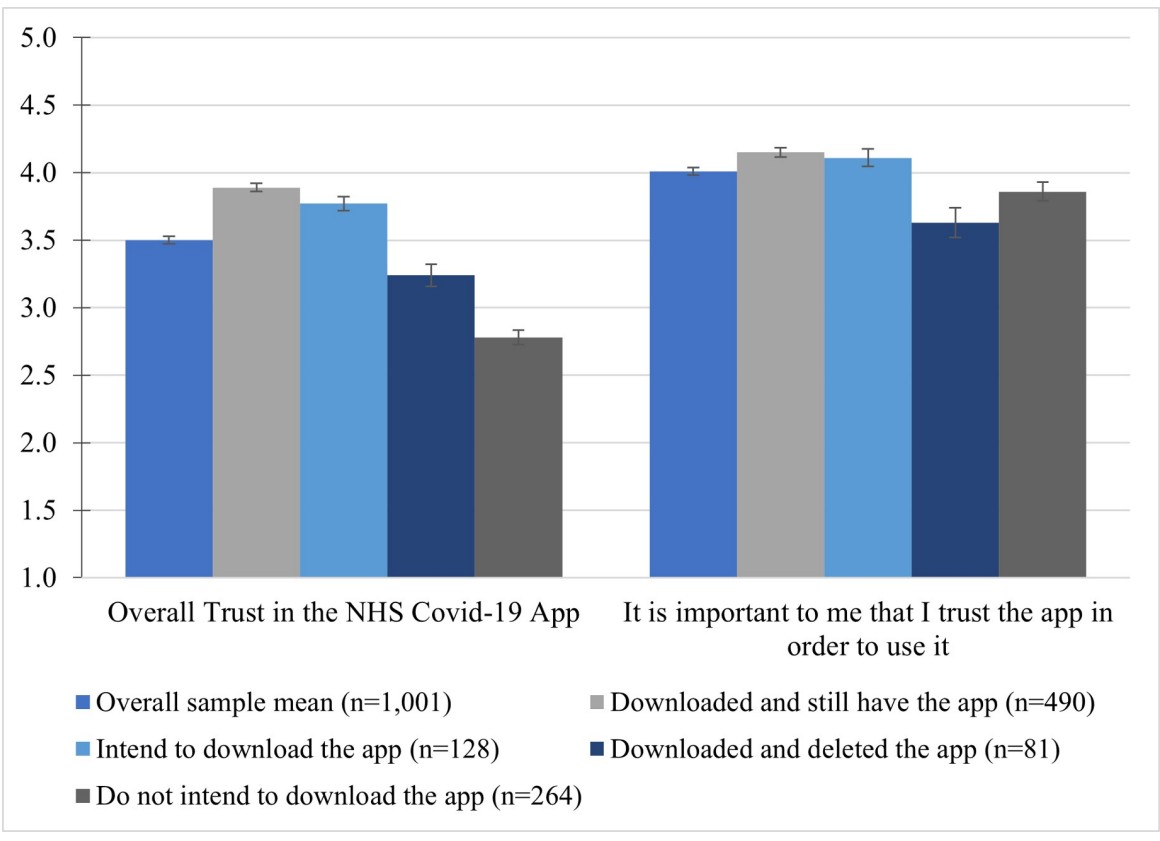

**Fig 3. Mean levels of trust and the important of trust in using the NHS COVID-19 app.** Mean scores with standard error bars.
1 = Strongly disagree, 2 = Somewhat disagree, 3 = Neither agree nor disagree, 4 = Somewhat agree, 5 = Strongly agree.

*p*<0.001). Pairwise comparisons with the Bonferroni correction for multiple tests shows that those who do not intend to download the app (mean = 2.78, SD = 0.85) have significantly lower trust than all other user groups, and those who downloaded it but went on to delete it (mean = 3.24, SD = 0.74) also have significantly lower trust than those with the app (mean = 3.89, SD = 0.96) or who intend to download it (mean = 3.77, SD = 0.58). There was no significant difference between those who downloaded it already and those who intended to do so. Trust in the app was felt to be important across all groups of users (Fig 2, mean = 4.01, SD = 0.029), An Independent-Samples Kruskal-Wallis test shows a significant difference between importance of trust among the four groups of participants ($X^2$(3) = 23.95, *p*<0.001). Pairwise comparisons with the Bonferroni correction for multiple tests shows that those who downloaded but deleted the app (mean = 3.24, SD = 0.98) place less importance on trust than those who still have the app (mean = 3.89, SD = 0.74), those who intend to download the app (mean = 4.11, SD = 0.72), and those who do not intend to download the app (mean = 3.86, SD = 1.15). For a breakdown of differences between groups on individual trust items, see [7]. The importance of trust correlates moderately with overall trust in the app, $r_s$ = 0.401, *p*<0.001.

### Reasons for and against participation and how this relates to trust

All interview participants were generally in favour of downloading a contact-tracing app, although some had reservations or pre-requisites for using it, such as not wanting to be tracked: *"the fact that they would know like just where I have been, I suppose it's not very pleasant thought"* (Participant L). Another reason not to use the app was the potential for being perceived as having done something 'bad' if they got traced: *"there could be a perception that they've been reckless, or that they are at fault or that they put your safety at risk."* (Participant J). Some preferred other ways of tracking their exposure: *"I'm basically going to do as much as I can on my own and not really trust what's happening outside"* (Participant A). Participants were also concerned about what being told to self-isolate could mean for people working in certain industries such as tourism and/or being self-employed: *"You know, weighing up in their minds whether or not the risk of coronavirus is higher than the risk of loss of salary"* (Participant I).

As shown above, those who did not download the app (n = 264) had lower overall trust in it but still placed the same importance on trust as other participants. Reasons for not downloading the app for questionnaire respondents re-iterated the concerns of the interview participants; mainly not wanting to be tracked (39.8%, n = 105), not thinking it would be effective (30.3%, n = 80), not wanting to take part in contact-tracing in that way (29.5%, n = 78), and a lack of trust in those who built the app (27.3%, n = 72).

Interview participants felt that framing the use of the app as helping others was important, *"if it's framed as in if you use this app, you're helping people. Rather than or protecting yourself, rather than just we'll notify you if you need to self-isolate,"* (Participant G), emphasising the need to think of others first, *"we are thinking of other people rather than ourselves I would hope."* (Participant H). Using the contact-tracing app was furthermore perceived as a way to overcome the pandemic: *"it sounds sensible and if people keep to it and adhere to it, it sounds like it's a good way to stop outbreaks"* (Participant D). In the same way, among questionnaire participants who intended to download the app (n = 128), their reasons were mostly wanting to help the NHS (65.6%, n = 84) or to help protect friends and family (62.5%, n = 80) or oneself (54.7%, n = 70), as well as reducing the spread of the virus (43.0%, n = 55) and helping to protect broader society (35.9%, n = 46). Participants who had downloaded the app (n = 490) were asked to indicate the extent to which they agreed with each reason; the strongest reasons were

also helping the NHS (mean = 4.42, SD = 0.75) and protecting friends and family (mean = 4.36, SD = 0.79). As shown in Table 3, reasons for downloading the app are stronger the higher the level of trust in the app. There are moderate correlations between trust and wishing to help the NHS, to reduce the spread of the virus, and to protect people, including the self, friends and family, and broader society; there are weak correlations between trust and downloading the app because everyone else was or because the government told them to. The remaining reasons for downloading show no relationship to trust. In terms of the relationship of motivations to the importance of trust, there is a moderate correlation between importance and the desire to protect oneself, and weak correlations between importance and helping the NHS, protecting friends and family, protecting broader society, and helping to reduce the spread of the virus.

## Trust in institutions related to Test and Trace

During the interviews, the NHS was generally perceived as the driving factor to prevent an even worse situation: *"So I think in general, if we were to have a major second spike, I think possibly wouldn't be as bad as it was, but that's not down to the government I think it's down to the NHS"* (Participant J). Participants did not express the same measure of trust when talking about the UK government, especially regarding data protection: *"the government has shown itself to be untrustworthy with personal information. [. . .] It feels like they might have a nefarious purpose for it"* (Participant J), calling for more transparency into how the government would be involved in a contact-tracing app.

When it comes to the big technology companies, a certain level of apathy was observable, along with the hope that they would overcome the need to compete with one another in favour of working for the good of everyone. *"I don't have a strong feeling, I just hope that the two of them [Apple and Google] are able to talk to each other and just make it work rather than go well this is my company and I'm Apple and I don't like anything that's not Apple kind of thing."* (Participant C).

Participants in the questionnaire also displayed strong levels of trust in the NHS (Table 4) and less in the UK government; trust in private contractors is weakest. A related-samples Friedman's two-way ANOVA shows a significant difference between trust in different institutions, $\chi^2(6) = 1191.982$, $p<0.001$. Post-hoc comparisons with the Bonferroni correction for

**Table 3. Reasons for downloading the NHS COVID-19 app, and their relationship to trust.**

| | Motivation for downloading app (n = 490) [a] | Correlation to Trust in the NHS COVID-19 app (n = 490)[b] | Correlation to the Importance of Trust in using the app (n = 490)[b] |
|---|---|---|---|
| **To help the NHS** | 4.42 (0.75) | 0.43*** | 0.29*** |
| **To help protect my friends and family** | 4.36 (0.79) | 0.47*** | 0.38*** |
| **To help protect myself** | 4.27 (0.88) | 0.47*** | 0.40*** |
| **To help protect broader society** | 4.20 (0.90) | 0.44*** | 0.34*** |
| **Because it will reduce the spread of the virus** | 4.11 (0.97) | 0.48*** | 0.37*** |
| **Because I need it to check into venues** | 3.54 (1.18) | 0.17*** | 0.08 |
| **Because the government told me to** | 3.46 (1.18) | 0.30*** | 0.12** |
| **Because everyone else is** | 3.14 (1.18) | 0.31*** | 0.09* |
| **Because it is a requirement for my job** | 2.53 (1.36) | 0.15*** | -0.04 |

[a] Those with the app (n = 490) were asked the extent to which they agreed that each reason was a motivation for downloading, 1 = Strongly disagree, 2 = Somewhat disagree, 3 = Neither agree nor disagree, 4 = Somewhat agree, 5 = Strongly agree. Mean score, standard deviation given in brackets.

[b] Spearman's rho correlations, significant correlations are flagged by *$p<0.05$, **$p<0.01$, or ***$p<0.001$

multiple tests show that trust in the NHS is significantly stronger than any other institution (all $p<0.001$), and trust in private contractors is significantly lower than any other institution (all $p<0.001$). Additionally, trust in the Big Tech companies is significantly lower than in small venues ($p<0.001$) and trust in the government is significantly lower than in small and large venues and local councils (all $p<0.001$); there are no other significant differences between groups.

The more users trust any of the institutions involved in the Test and Trace system, the greater their trust in the app (Table 4). Levels of trust are moderately related to trust in the UK government, local government, big tech companies, private contractors, the NHS, and large hospitality venues, but only weakly related to small hospitality venues. Additionally, the more importance users place on trust, the higher their trust in institutions; however there is only a weak relationship between importance and trust in the NHS, and no relationship with the remaining institutions.

## Trust and the technology used by the app

Interview participants stressed that being able to verify information from an app was highly important. This, as one participant pointed out, might stem from learnt distrust towards outside contacts on their phone or other forms of communication, *"You see I think these days we've all got so used to not trusting anybody who rings us. You know, it's sort of drummed into you."* (Participant H). The need for verification started as early as downloading, to make sure that it was clear which app was the official software, for example by making it available through an official *"link through to the Apple store website, so yeah, something rather than just searching for it on the App Store"* (Participant J). Being verified also meant an increase in trust in the system for participants *"I'd trust a system either way if it was if it, you know if it felt legitimate and authentic and it was, it was able to identify itself properly"* (Participant A).

Some interview participants stressed that being able to talk to a human rather than having to rely on an automated system was important. *"I appreciate that one of the big benefits with that is you know you can develop a relationship with someone there, which I'd expect it would be quite important if you're asking someone to lock themselves in a room for the next couple of weeks"* (Participant A), which could be integrated into a semi-automated system as well. *"So, I think there's maybe, uh, maybe the fact that if you had an app then you could have a follow up phone call if you could, if you requested it on the app so that you can get a bit of reassurance and know exactly what's going on from a human being."* (Participant I). There was furthermore

**Table 4. Trust in groups involved in the Test and Trace system, and their relationship to trust in the app.**

| I trust... | Trust[a] | Correlation to Trust in the NHS COVID-19 app[b] | Correlation to the Importance of Trust for use[b] |
|---|---|---|---|
| **The NHS** | 4.11 (0.95) | 0.47*** | 0.25*** |
| **Small hospitality venues, such as independent pubs and cafes** | 3.44 (0.99) | 0.34*** | 0.13*** |
| **Larger hospitality venues, such as chain restaurants** | 3.32 (1.01) | 0.41*** | 0.13*** |
| **My local council** | 3.30 (1.06) | 0.51*** | 0.19*** |
| **The big tech companies, such as Google and Apple** | 3.18 (1.11) | 0.50*** | 0.15*** |
| **The UK Government** | 3.01 (1.28) | 0.53*** | 0.14*** |
| **Private contractors, such as Serco** | 2.83 (1.10) | 0.49*** | 0.08* |

[a] n = 1,001. Mean scores followed by standard deviations in brackets. 1 = Strongly disagree, 2 = Somewhat disagree, 3 = Neither agree nor disagree, 4 = Somewhat agree, 5 = Strongly agree

[b] Spearman's rho correlations, significant correlations are flagged by *$p<0.05$, **$p<0.01$, or ***$p<0.001$

some expression of trusting humans more than machines to consider the available evidence, *"Having this person tell you to self-isolate is much more, has much more of an impact in my opinion, because in your mind, you're thinking OK, well, they do. The person must have searched. They must have checked everything"* (Participant L).

Nevertheless, participants were also aware of the limiting factor a greater number of humans in the system may have, including issues of time and capacity: *"having as much time as possible to take action for my friends and family and others around is more important to me than a human ringing me."* (Participant C). In addition, there is also the potential for bias that comes with human involvement: *"different studies have revealed different things, so I think that probably automation would be a safer way to go in terms of the blame game to this."* (Participant G), and the idea that it might lead to unequal treatment between people. *"I think the hardest part is the inconsistency of different places have got different things, so yes, if there was an app and it kind of tracks and traces you automatically, then that seems quite sensible."* (Participant D).

Relatedly, questionnaire participants who had the app, intended to have the app, or had deleted the app (n = 699) were asked how they believed the app makes decisions. Most (falsely) believed that it is a combination of human decision-making and being automated by the app (54.2%, n = 379). Only 19.5% (n = 136) of those surveyed rightly believed that decisions making was entirely automated; a quarter (26.3%, n = 184) believed that decisions were made entirely by humans. However, participants tended to agree that they understood how the NHS COVID-19 app worked, and higher trust in the app is moderately related to a higher feeling of understanding and weakly to stronger feelings that it is important to trust the app (Table 5).

Table 5 shows how questions relating to the technology and the features of the app relate to trust in the app among questionnaire participants who had the app, intended to have the app, or deleted the app (n = 699). As in the interviews, they agreed that it was it was important that the app provided explanations for information given to them, that they could verify that notifications were authentic, and that they could speak to a person about any advice from the app;

**Table 5. Agreement with statements related to the technology and ecosystem surrounding the NHS COVID-19 app[a], and their relationship with trust in the app[b].**

| | Mean (SD) | Correlation to Trust in the NHS COVID-19 app | Correlation to the Importance of Trust for use[b] |
|---|---|---|---|
| **It is important to me that I can verify that notifications from the app are authentic** | 4.08 (0.89) | 0.30*** | 0.40*** |
| **The app is easy to use (n = 571)[c]** | 4.07 (0.90) | 0.48*** | 0.37*** |
| **The app is useful to wider society** | 4.02 (0.90) | 0.57*** | 0.39*** |
| **It is important to me that I can get an explanation for any information given to me by the app** | 3.98 (0.85) | 0.26*** | 0.34*** |
| **I understand how the NHS COVID-19 app works** | 3.90 (0.88) | 0.47*** | 0.34*** |
| **The app is useful to me personally** | 3.75 (0.97) | 0.60*** | 0.38*** |
| **It is important to me to be able to speak to a person about any advice given by the app** | 3.75 (0.97) | 0.25*** | 0.24*** |
| **The regulations governing the creation of the app are sufficient** | 3.64 (0.94) | 0.53*** | 0.28*** |
| **It is important to me that I can opt-in and opt-out of contact-tracing** | 3.57 (1.11) | 0.14*** | 0.12** |
| **I am concerned about how my data will be used by the app** | 3.18 (1.3) | -0.18*** | -0.07 |
| **I felt that I had no choice but to download the app (n = 571)[c]** | 3.02 (1.26) | -0.01 | -0.04 |
| **I have felt frustrated as a result of a notification from the app (n = 571)[c]** | 2.69 (1.28) | -0.05 | -0.15*** |

[a] n = 699. Standard deviations are given in brackets following the mean. 1 = Strongly disagree, 2 = Somewhat disagree, 3 = Neither agree nor disagree, 4 = Somewhat agree, 5 = Strongly agree

[b] Spearman's rho correlations, significant correlations are flagged by *$p < 0.05$, **$p < 0.01$, or ***$p < 0.001$

[c] n = 571 because these statements were only asked to participants who had used the app

these factors also moderately related to trust in the app. They also tended to agree that the app was useful to them and wider society, and easy to use; these factors were also moderately related to trust. They agreed less that the regulations that govern the app were sufficient, which had a moderate relationship with trust, and that it was important that they could opt-in and out of contact-tracing, which showed no relationship to trust. They were neutral about their concern about the use of data by the app and tended to disagree that they had no choice but to download the app or that they had been frustrated by their use of the app, none of which displayed a relationship to trust. In terms of the importance of trust in decisions to use the app, this was moderately correlated with the importance of verifications, and weakly correlated to feelings that the app was easy to use, that it was useful to the user and to wider society, the importance of explanations, understanding, and the feelings that the regulations governing the creation of the app were sufficient.

## Discussion

Herein we discuss our findings regarding trust, (non-)adoption and the issues it raises for digital-contact-tracing and its wider societal consequences.

### Trust is related to both adoption and non-adoption of the app

The results show that trust has a significant effect on whether people adopt (i.e., download) the app or not. To reiterate the main findings from the questionnaire study, trust in the app is significantly lower in people who chose not to download the app, and those who deleted the app after download. Only half of the participants reported they had downloaded the app, and whilst another 13% intended to download it, results suggest that trust affects adoption as also suggested by others [14, 36]. Despite previous studies suggesting that acceptance is high [10–12, 14, 26] uptake in this study is similar to that reported for the UK at large [27, 28], and these low levels of uptake may be concerning given the WHO's recommendation to trace 80% of contacts within 3 days [1]. Following our results, for digital contact-tracing to be effective, in the UK at least, it needs to be supplemented with other means, such as 'manual', human-led tracing to reach the recommended levels of uptake.

Trust is significantly correlated with various motivations for downloading the app and various features of the app itself, which therefore may affect adoption. A desire to help protect people (the NHS, self, friends and family, wider society) was related to higher trust, as discussed in the following section. Non-adoption of the app or deleting it related to a lack of trust in the app tracking users, preferring other ways of carrying out contact-tracing and dealing with the pandemic, a belief that the app is ineffective, and a lack of trust in the people who made the app. Other studies have also related this dislike of being tracked and concern for privacy to a lack of trust [14, 15, 36], and TAM highlights believing in the usefulness (or efficacy) of technology is a factor in its adoption [17].

Trust in the apps' use of data was neutral and not related to overall trust in the app or how important trust was to the user; additionally concerns about data use by the app were overridden by the need for protection, leading to people to use the app even when these concerns existed. In fact, participants tended to have higher trust in the use of data and the app working as it should than in the behaviour of other people. There is some evidence then, that privacy concerns are put aside in the context of a pandemic [20], perhaps in trading for the greater good of public health; however, we would not go as far as to suggest that privacy is not important per se; indeed, research has shown that privacy was among the most debated topics about the German contact-tracing app [40]. Additionally, being able to verify and get explanations

for what the app was doing was felt to be important and this showed a relationship to both actual trust and the importance of trust.

Those with higher trust in the app overall also agreed more strongly that they understood how the app worked; however, the results show that understanding was actually very low, with many people believing that humans were still involved in decisions to alert users to self-isolate, when in fact, this decision is entirely automated [5]. There is perhaps an element of 'wishful thinking', with respondents stating that human contact was felt to be important, in the form of being able to speak to a person about what the app told them, and this was significantly related to trust. Our findings echo those of Lu et al., who found in a survey of 291 US-based respondents that digital contact-tracing was perceived more beneficial for protecting privacy and providing convenience and accuracy, while the human approach could provide emotional assurance and advice [21].

Positively, participants felt that the app was useful, reliable, and easy-to-use; these are important factors in the Technology Acceptance Model [17], and higher usefulness has previously been shown to relate to motivations for social distancing and contact-tracing app use [19]. Usefulness (to self and society) and ease of use was related to higher trust overall in the app, as well as how important trust was felt to be. This suggests that for technology, trust is related to the user-experience and therefore willingness to use it. The choice to opt-in and -out of contact-tracing was not seen as important and was not related to trust; whether respondents felt they had a choice in downloading the app and whether they experienced frustration also did not relate to trust. This implies that voluntariness and choice did not clearly relate to either adoption or trust in the app.

## Trust in digital contact-tracing is embedded in wider societal concerns

Beyond trust related to understanding and attitudes towards the app, our results show how adoption and rejection of digital contact-tracing solutions can be understood more broadly, considering the wider social and societal context. The views towards family, loved ones, and other members of society matter as well as views towards stakeholders and institutions such as the NHS, the government, and private companies involved in contact-tracing. Lack of trust has been found to be a significant barrier to acceptance of both hypothetical [10] and existing digital contact-tracing [19]. As presumed from other studies, that higher levels of trust in the government and health authorities lead to increased uptake of such apps [17], our results also show that many concerns relating to trust and other factors relate to broader influences on trust in society. Beliefs held prior to the release of the app, such as concerns about the intentions of official bodies, were transferred to intention to use the app. It is important to understand the context into which a technology is implemented to help public health providers avoid potential issues when releasing interventions to the general public, and promote uptake on release.

Results surrounding motivations for downloading and using the app indicate stronger motivation for downloading the app correlates with higher trust in the app overall, which are especially related to keeping oneself and loved ones safe from the pandemic. This is shown particularly by the correlations between trust and the motivations of wishing to help the NHS, reduce the spread of the virus, and protecting people. Other studies have shown how protecting others is important for acceptance of contact-tracing [10, 13, 14], likewise here it is also shown to be important for trust. Prior work has argued that altruism is an important factor underpinning participation in activities that promote public health and support the NHS, such as Titmuss' account on blood donation from the 1970s [41]. Related to the COVID-19 pandemic, Lucivero et al. have also found a range of normative positions in their interviewees'

accounts, with positive-leaning accounts invoking notions of the public or 'greater' good and relieving stress on the health care system in their reasoning [42]. It may be that the desire to help protect people (the NHS, self, friends and family, wider society) also increases the motivation to trust the technology, and in turn trust is achieved [35]. These factors also related to an increased sense of the importance of trust. Trust was not related to positive or negative feelings of being forced to comply, shown by a lack of relationship between trust and downloading the app due to this being socially or lawfully enforced. However, whilst compliance was not a strong motivator, the interview data clarified that participants understood that it was important that as many people as possible take part in contact tracing. Practical reasons for using the app (for example requirements for daily life) also did not appear to affect levels of trust.

In general terms, the NHS is trusted over the government, and there is a higher level of trust towards devolved governments of the UK (e.g., Scotland), but the main issue for the UK public was their lack of belief in other people to 'do the right thing'. It is extremely important when designing technology for uptake of a wide audience to consider not just the user experience of the technology but also the wider social and societal context surrounding it. Our results show that trust and adoption of the app is not just about the app itself, the issues are much broader.

Trust in the app is substantially related to trust in the surrounding institutions, which therefore is likely to influence adoption. This study shows that the NHS is trusted significantly more than any other institution, which is positive considering that the perceived effectiveness and usefulness of contact-tracing apps may depend on the health system in which it is embedded [18]. However, whilst higher trust in the app was linked to having higher trust in all the institutions studied, it was particularly strongly related to trust in the government, local councils, and big technology. These all have relatively low levels of trust among users, which in turn leads to low trust in the app. Previous research has also found distrust in the government relates to lower acceptance [15, 16], but whilst in this study trust in the government was low, and may therefore contribute to non-use of the app, the regulations surrounding the creation of the app were felt to be acceptable, and this was moderately related to trust so may to some extent allay concerns. Trust in the government may be eroded by other factors such as reports of their actions in the media, rather than the embedded laws and regulations that they oversee.

## What it means for the design of digital contact-tracing technologies

There are many lessons that can be learnt from the study of the digital contact tracing solutions for the COVID-19 pandemic that can be taken forward in future designs of large-scale public health technologies. These lessons go beyond understanding the specific technology in use or features of an app related to the COVID pandemic, but also to wider understanding about the importance of societal issues surrounding use of technologies implemented by societal bodies. Echoing the need for a wider understanding, a study that examined attitudes towards digital contact tracing in the main German-speaking countries identified trust in authorities, respect of individual privacy, voluntariness, and temporary use of contact tracing apps as prerequisites for democratic compatibility [43]. Our studies show the critical importance of trust in relation to (non-)participation in digital contact-tracing; studies of technology use and acceptance should investigate the role of trust in their respective domains. Not only because the popular TAM2 and its successor, the Unified theory of acceptance and use of technology (UTAUT) do not include trust by default [17, 44], but also because, especially in pervasive topics such as the pandemic that touch many aspect of people's lives, issues of trust are more far-reaching than views towards the technology itself. Trust and distrust in other members of society and institutions at the centre of the pandemic, for instance, can become a barrier to the adoption of

technology supposed to help reduce the spread. Furthermore, there is evidence that (dis-)trust in institutions is not equally distributed among citizens, with people self-reporting as belonging to black, Asian, or other ethnic minorities reporting less trust in the NHS [7]. Thus, addressing these trust issues requires a nuanced approach, beginning by understanding in more depth the concerns different communities have regarding contact-tracing.

Designing technology such as digital contact-tracing, which is for wide-scale use and has potentially severe consequences for a population, needs a holistic approach. It is not necessary to dismiss short-term technical improvements of the App or marketing/educational campaigns, which have been shown to a positive effect in a public health crisis [45]. Prior work that has examined attitudes towards contract-tracing has for instance suggested that a hybrid approach combining the strengths of human and automated tracking may be more humane while preserving privacy [21, 22]. Another study examining the uptake of the digital contact-tracing solution in Germany has shown that while informational messaging has a limited effect on uptake, monetary incentives can strongly increase uptake [46].

However, it would be naïve to suggest that technical solutions and campaigns are a panacea. It is essential to consider long-term mechanisms that have proved to be effective at involving the public in health research and development, and how these may be leveraged to increase public trust in technological initiatives aimed at improving public health. One such approach that has proved effective in the UK context, similar in ethos to participatory and co-design, is Patient and Public Involvement (PPI) [47]. PPI involves actively working with members of the public and patients in health research, with "Research being carried out 'with' or 'by' members of the public rather than 'to', 'about' or 'for' them" [48], and positive impacts including enhanced quality and appropriateness of research have been for all stages of health research [49]. PPI has for instance been shown to be successful in increasing enrolment in clinical trials [50], understanding how to use 'wearables' in health research for dementia [51], and in the design of assistive technology devices [52]. There have also been suggestions that involving PPI groups in COVID-19 policy making would have been beneficial [53]. A further notable example is the Responsible Research and Innovation framework (RRI), which advocates engaging users in the development, implementation and evaluation; this includes considering the wider social and societal implications of how and why a technology might be accepted or rejected by the public [54, 55]. Meaningful participation is difficult and costly, but methods such as using representation artefacts can avoid 'tokenistic participation' [56]. Such initiatives should be embedded in the design of technology, addressing the barriers to adoption early and more deeply to understand what needs to be done to improve trust in key stakeholders at the centre of public health.

## Limitations

The caveat for the qualitative thematic analysis and the insights it provides into the broader societal system into which the app was released is that the interviews were conducted with a small sample of the UK public before the NHS COVID-19 app was released, therefore these results are based on hypothetical contact tracing. However, we believe there is value in relating what people imagine will be the case to how they felt when the app was actually released. Future work should look at how these attitudes towards the real app have changed over the previous year, and with prolonged use of the app. Additionally, whilst the sample was small, a good level of saturation in responses was reached.

Whilst care was taken to ensure that the questionnaire sample was representative of the UK population in terms of age, gender, region, and ethnicity, other factors such as income and political leaning may have affected the results. Future work should also consider multivariable

analyses to account for the demographics of participants, to examine other differences in trust, for example lower trust in the government or the NHS might be driven by factors such as age, gender, or education.

The sample was also taken from an online panel, which creates a bias towards the online population; however, for this study this was considered acceptable due to the focus on a smartphone application which assumes internet use. Whilst approximately 92% of adults in the UK are recent internet users, and only 6.3% of adults had never used the internet [57], this does mean that this may have excluded some members of the UK public from the study, who may be disproportionately affected by the pandemic.

Neither of the studies in this paper looked at how much participants engaged with the app itself, beyond the fact that they downloaded it, and questionnaire participants were asked whether (but not how much) they had used various features. This would have been beneficial to understand more deeply the reasons for their responses related to trust in the app. This paper also has not reported much about the reported compliance to the app and notifications to self-isolate, as these were already reported in [7]. However, in future work it would be beneficial to provide context around reported behaviour in response to the app.

Finally, it is often the case that trust, whether it be in technology, institutions, or other people, is often underpinned by other psychological factors such as a person's tolerance of uncertainty and attachment style, as well as levels of authority, and locus of control. It would have been useful to have included measures of such constructs in the study to understand how these relate to the feelings of trust (or not) of participants, and which factors explain the experience of trust in the app.

## Conclusion

The paper presented research into how users felt about the Test and Trace ecosystem in the UK and their overall trust in the NHS COVID-19 smartphone app, the digital contact-tracing solution in the UK. It investigated how trust in the app relates to reasons for and against participation in digital contact-tracing, by proxy of downloading the app, trust in the institutions and groups related to Test and Trace, and opinions about the technology and features employed by the app. Results show that trust is related to adoption and non-adoption of the app, but more broadly, trust in digital contact-tracing is embedded in wider social and societal concerns about family and loved ones and trust in institutions at the centre of the pandemic response.

Those who downloaded the app have significantly higher trust in it overall than those who do not intend to download it or have deleted it; a lack of trust in those who built the app was one of the most common reasons for not downloading the app. The strongest reasons for downloading were helping oneself and others, including the NHS, friends and family, and wider society. These factors were all positively related to levels of trust and feeling that trust is important in using the app. App users generally agreed that the app was easy to use and useful, which also positively related to their trust in it. Respondents also agreed that it was important that the app provided explanations and verification of information and notifications from the app, and that they could speak to a person about advice given; higher agreement to all these factors were related to higher trust in the app. Additionally, the more users trust the institutions involved in contact-tracing; trust was highest in the NHS but trust in the app was particularly related to trust in the government, local councils, and big technology companies, which have relatively low levels of trust among users. Whilst the COVID-19 pandemic appears to be on the way out, this paper gives insight into how members of the public may react to future interventions designed to tackle issues of public health. The lessons learnt are transferrable to

future situations. Trust should be considered deeply in technology adoption research, including participatory approaches which engage communities in the design of technology and work to understand barriers to adoption beyond the features of the technology itself, to contribute to a more trusted ecosystem of digital contact-tracing. For the future, lessons gained from studying digital contact tracing should be used in the design and implementation of other large scale public health technologies, to ensure trustworthiness and wider acceptance and use.

## Author Contributions

**Conceptualization:** Liz Dowthwaite, Hanne Gesine Wagner, Camilla May Babbage, Joel E. Fischer, Pepita Barnard, Elena Nichele, Elvira Perez Vallejos, Jeremie Clos, Virginia Portillo, Derek McAuley.

**Data curation:** Liz Dowthwaite, Hanne Gesine Wagner, Camilla May Babbage, Joel E. Fischer.

**Formal analysis:** Liz Dowthwaite, Hanne Gesine Wagner, Camilla May Babbage.

**Funding acquisition:** Joel E. Fischer, Derek McAuley.

**Investigation:** Liz Dowthwaite.

**Methodology:** Liz Dowthwaite, Hanne Gesine Wagner, Camilla May Babbage, Joel E. Fischer, Elvira Perez Vallejos, Virginia Portillo.

**Project administration:** Joel E. Fischer.

**Supervision:** Joel E. Fischer.

**Validation:** Liz Dowthwaite, Hanne Gesine Wagner, Camilla May Babbage, Pepita Barnard, Elena Nichele.

**Writing – original draft:** Liz Dowthwaite.

**Writing – review & editing:** Liz Dowthwaite, Hanne Gesine Wagner, Camilla May Babbage, Joel E. Fischer, Pepita Barnard, Elena Nichele, Elvira Perez Vallejos, Jeremie Clos, Virginia Portillo.

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
