## [Decision Letter · Decision Letter 0]

8 Aug 2022

PONE-D-22-17800The Relationship between Trust and Attitudes towards the COVID-19 Digital Contact-Tracing App in the UKPLOS ONE

Dear Dr. Dowthwaite,

Thank you for submitting your manuscript to PLOS ONE. After careful consideration, we feel that it has merit but does not fully meet PLOS ONE’s publication criteria as it currently stands. Therefore, we invite you to submit a revised version of the manuscript that addresses the points raised during the review process.

 Please revise the manuscript as suggested by the reviewers. Please submit your revised manuscript by Sep 22 2022 11:59PM. If you will need more time than this to complete your revisions, please reply to this message or contact the journal office at plosone@plos.org. Please include the following items when submitting your revised manuscript:

We look forward to receiving your revised manuscript.

Kind regards,

Mukhtiar Baig, Ph.D.

Academic Editor

PLOS ONE

Journal Requirements:

2. In the ethics statement in the Methods, you have specified that verbal consent was obtained. Please provide additional details regarding how this consent was documented and witnessed, and state whether this was approved by the IRB.

3. You indicated that you had ethical approval for your interview study. Please clarify whether minors were involved in the interviews. If so, in your Methods section, please ensure you have also stated whether you obtained consent from parents or guardians of the minors included in the study or whether the research ethics committee or IRB specifically waived the need for their consent.

4. You indicated that you had ethical approval for your online survey study. In your Methods section, please ensure you have also stated whether you obtained consent from parents or guardians of the minors included in the study or whether the research ethics committee or IRB specifically waived the need for their consent.

6. Please include a caption for figure 2. 

Additional Editor Comments:

Following are a few other observations sent by the third reviewer via email. Please reply to these comments as well.

• Please write a clear justification for the work

• No clear research question

• No structure to the abstract

• Chunks of the text in the Introduction belong in the Discussion

• It needs a restructure

• It also seems quite late in the day to be covering this topic

• I’m not sure it adds significantly to the already published work

• Referring to the attached paper might help the authors – it’s not in their reference list

Jones K, Thompson R. To Use or Not to Use a COVID-19 Contact Tracing App: Mixed Methods Survey in Wales. JMIR mHealth and uHealth. 2021 Nov 22;9(11):e29181.

Reviewers' comments:

Reviewer's Responses to Questions

**Comments to the Author**

1. Is the manuscript technically sound, and do the data support the conclusions?

Reviewer #1: Yes

Reviewer #2: Partly

2. Has the statistical analysis been performed appropriately and rigorously? 

Reviewer #1: Yes

Reviewer #2: I Don't Know

3. Have the authors made all data underlying the findings in their manuscript fully available?

Reviewer #1: Yes

Reviewer #2: Yes

4. Is the manuscript presented in an intelligible fashion and written in standard English?

Reviewer #1: Yes

Reviewer #2: Yes

5. Review Comments to the Author

Reviewer #1: This is a strong paper with some useful findings, and one which I read with interest as someone with similar data I am working on myself. I can therefore see that this is a well conducted piece and that the findings are useful, and with application to other public health situations in future, beyond Covid, potentially. The measures are sound, the analysis correct, and the writing clear. I have just one suggestion, which might be useful to make reference to in the latter part of the paper. It would have been useful to recognise that trust in apps, the government, and things in general, are usually underpinned by various psychological variables such as intolerance of uncertainty, authoritarianism, locus of control and attachment style. So, it would have been useful to measure one or more of these, and then partial out those variables in order to feel more sure that the findings are not polluted by such factors. It's not essential for publication, but I think it might strengthen the limitations/conclusions to recognise this.

Reviewer #2: Thank you for asking me to review this interesting paper on public trust in the COVID-19 contact tracing apps. In this paper, the authors draw on semi-structured interviews with 12 people prior to the introduction of a contact tracing app in the UK, and a large scale questionnaire to highlight a number of key points. Most interesting (and novel at least in my opinion) are points 2 and 4 (noted on page 3 of the manuscript).

I think there are some issues that the authors could address to further strengthen their paper. I have note these below.

I find it hard to marry the two data sets and I think more work is needed to make it clear that the combination is meaningful. 12 semi-structured interviews prior to seeing a contact tracing app (and reflecting on their use in other countries) seems rather divorced from data once a formal contact tracing app had been introduced and means the comparison is between the hypothetical and the real.

In the discussion of trust, I think it’s important to also note and learn from accounts of civic participation (in health), accounts of altruism and participation in other activities that supporting the NHS, for example, blood donation (and rationales for doing so). Titmuss’s work on blood is donation is a good starting point here (reference from a reprint - it was originally published in the 70s) Titmuss, R. (2018). The Gift Relationship: From Human Blood to Social Policy. Bristol University Press.

Linked to these points about civic participation, there is also substantial work on the concept of solidarity which appears to me as something that should be noted and could help develop your discussion (particularly in relation to key findings 2 and 4.). Barbara Prainsack’s work is crucial here. She was also involved in a large pan Europe project on covid-19 with two papers that are linked to covid-19 apps that are currently not covered. These papers might help further develop your discussion of your findings.

Federica Lucivero, Luca Marelli, Nora Hangel, Bettina Maria Zimmermann, Barbara Prainsack, Ilaria Galasso, Ruth Horn, Katharina Kieslich, Marjolein Lanzing, Elisa Lievevrouw, Fernandos Ongolly, Gabrielle Samuel, Tamar Sharon, Lotje Siffels, Emma Stendahl & Ine Van Hoyweghen (2022) Normative positions towards COVID-19 contact-tracing apps: findings from a large-scale qualitative study in nine European countries, Critical Public Health, 32:1, 5-18, DOI: 10.1080/09581596.2021.1925634

Zimmermann BM, Fiske A, Prainsack B, Hangel N, McLennan S, Buyx A Early Perceptions of COVID-19 Contact Tracing Apps in German-Speaking Countries: Comparative Mixed Methods Study J Med Internet Res 2021;23(2):e25525

doi: 10.2196/25525

Finally, I’m a little perplexed by finding 1 (from page 3). To me this appears pretty obvious and I wonder whether there is more nuance that could be teased out from the data. For example, does the data allow the authors to say anything about other rationales for adopting, adopting and abandoning or avoiding the Contact Trace App? Is trust the highest scoring rationale (in relation to other rationales - like civic duty, trust in the NHS as opposed to the app etc and could these also be noted so it’s clear that the finding is important and contributes meaningfully to our knowledge.

Data Availability: I've noted that data has been made available above (which I realise is incorrect based on the authors' statements) - this is just to note that the rationale for not making qualitative data available is sound to me (and common practice).

6. PLOS authors have the option to publish the peer review history of their article (what does this mean?). If published, this will include your full peer review and any attached files.

Reviewer #1: No

Reviewer #2: No

---

## [Author Response · Author response to Decision Letter 0]

28 Sep 2022

Please ensure that your manuscript meets PLOS ONE's style requirements, including those for file naming. The PLOS ONE style templates can be found at https://journals.plos.org/plosone /s/file?id=wjVg/ PLOSOne_formatting_sample_main_body.pdf and https://journals.plos.org/plosone /s/file?id=ba62/ PLOSOne_formatting_sample_title_authors_affiliations.pdf - Files renamed as requested

In the ethics statement in the Methods, you have specified that verbal consent was obtained. Please provide additional details regarding how this consent was documented and witnessed, and state whether this was approved by the IRB. - Clarified that this was in addition to the online consent form, and was approved

You indicated that you had ethical approval for your interview study. Please clarify whether minors were involved in the interviews. If so, in your Methods section, please ensure you have also stated whether you obtained consent from parents or guardians of the minors included in the study or whether the research ethics committee or IRB specifically waived the need for their consent. - Clarified that no minors were involved in the study

You indicated that you had ethical approval for your online survey study. In your Methods section, please ensure you have also stated whether you obtained consent from parents or guardians of the minors included in the study or whether the research ethics committee or IRB specifically waived the need for their consent - Clarified that no minors under 16 were involved in the study, and those under 18 did not require additional consent

Please include your full ethics statement in the ‘Methods’ section of your manuscript file. In your statement, please include the full name of the IRB or ethics committee who approved or waived your study, as well as whether or not you obtained informed written or verbal consent. If consent was waived for your study, please include this information in your statement as well. - Information provided as requested

Please include a caption for figure 2. - Figure 2 was mislabelled as figure 1 in the manuscript

Reviewer 1 

This is a strong paper with some useful findings, and one which I read with interest as someone with similar data I am working on myself. I can therefore see that this is a well conducted piece and that the findings are useful, and with application to other public health situations in future, beyond Covid, potentially. The measures are sound, the analysis correct, and the writing clear. - We thank the reviewer for this response

I have just one suggestion, which might be useful to make reference to in the latter part of the paper. It would have been useful to recognise that trust in apps, the government, and things in general, are usually underpinned by various psychological variables such as intolerance of uncertainty, authoritarianism, locus of control and attachment style. So, it would have been useful to measure one or more of these, and then partial out those variables in order to feel more sure that the findings are not polluted by such factors. It's not essential for publication, but I think it might strengthen the limitations/conclusions to recognise this. - Thank you. We have added this consideration to the limitations of the paper.

Reviewer 2 

Thank you for asking me to review this interesting paper on public trust in the COVID-19 contact tracing apps. In this paper, the authors draw on semi-structured interviews with 12 people prior to the introduction of a contact tracing app in the UK, and a large scale questionnaire to highlight a number of key points. Most interesting (and novel at least in my opinion) are points 2 and 4 (noted on page 3 of the manuscript). - Thank you for this comment

I find it hard to marry the two data sets and I think more work is needed to make it clear that the combination is meaningful. 12 semi-structured interviews prior to seeing a contact tracing app (and reflecting on their use in other countries) seems rather divorced from data once a formal contact tracing app had been introduced and means the comparison is between the hypothetical and the real. - We have added further justification at strategic places throughout the text, and added to limitations. This includes clarifying that we are looking at how the hypothetical (ie how people felt about the idea of the app and how they thought they’d feel about it’s release) relates to the real (how they actually felt when it was used ‘in the wild’) – rather than a comparison of the two.

In the discussion of trust, I think it’s important to also note and learn from accounts of civic participation (in health), accounts of altruism and participation in other activities that supporting the NHS, for example, blood donation (and rationales for doing so). Titmuss’s work on blood is donation is a good starting point here (reference from a reprint - it was originally published in the 70s) Titmuss, R. (2018). The Gift Relationship: From Human Blood to Social Policy. Bristol University Press. - We thank the reviewer for this useful reference and have added it to the Discussion

Linked to these points about civic participation, there is also substantial work on the concept of solidarity which appears to me as something that should be noted and could help develop your discussion (particularly in relation to key findings 2 and 4.). Barbara Prainsack’s work is crucial here. She was also involved in a large pan Europe project on covid-19 with two papers that are linked to covid-19 apps that are currently not covered. These papers might help further develop your discussion of your findings.

- Federica Lucivero, Luca Marelli, Nora Hangel, Bettina Maria Zimmermann, Barbara Prainsack, Ilaria Galasso, Ruth Horn, Katharina Kieslich, Marjolein Lanzing, Elisa Lievevrouw, Fernandos Ongolly, Gabrielle Samuel, Tamar Sharon, Lotje Siffels, Emma Stendahl & Ine Van Hoyweghen (2022) Normative positions towards COVID-19 contact-tracing apps: findings from a large-scale qualitative study in nine European countries, Critical Public Health, 32:1, 5-18, DOI: 10.1080/09581596.2021.1925634

- Zimmermann BM, Fiske A, Prainsack B, Hangel N, McLennan S, Buyx A Early Perceptions of COVID-19 Contact Tracing Apps in German-Speaking Countries: Comparative Mixed Methods Study J Med Internet Res 2021;23(2):e25525

doi: 10.2196/25525 - Thank you for these points, we have added the suggested references to the Discussion

Finally, I’m a little perplexed by finding 1 (from page 3). To me this appears pretty obvious and I wonder whether there is more nuance that could be teased out from the data. For example, does the data allow the authors to say anything about other rationales for adopting, adopting and abandoning or avoiding the Contact Trace App? Is trust the highest scoring rationale (in relation to other rationales - like civic duty, trust in the NHS as opposed to the app etc and could these also be noted so it’s clear that the finding is important and contributes meaningfully to our knowledge. - We have demoted point 1 to the final point, focussing on the lack of trust as one of the strongest reasons for not downloading, so that the summary focuses on the most interesting points

Reviewer 3 

Please write a clear justification for the work - Edited the Introduction to make the aims clearer

No clear research question - Clarified the exploratory and descriptive nature of the paper

No structure to the abstract - We have edited the abstract to make it clearly structured as background, method, results and implications. There is no requirement from PLOS One to use a specific (headered) structure

Chunks of the text in the Introduction belong in the Discussion - Everything included in the Introduction was from previous studies rather than our own so we wanted to highlight them early on. We have added parts to the discussion which reflect back on these previous studies. 

It also seems quite late in the day to be covering this topic - Whilst this pandemic is on the way out, this paper gives insight into how members of the public may react to future interventions designed to tackle issues of public health. The lessons learnt are transferrable to future situations and we have made this clearer in the Discussion 

I’m not sure it adds significantly to the already published work - This paper includes a thematic analysis of data that has not appeared before, as well as many new tests of the relationship of trust to other factors. The previous paper did not look at the relationships between different factors at all. The inclusion of trust in the app as a scale to compare to other factors is entirely new. The vast majority of statistical analysis is new, but we had to re-report things like mean scores to contextualise these results.

Referring to the attached paper might help the authors – it’s not in their reference list Jones K, Thompson R. To Use or Not to Use a COVID-19 Contact Tracing App: Mixed Methods Survey in Wales. JMIR mHealth and uHealth. 2021 Nov 22;9(11):e29181. - Thank you for the addition of this useful reference.

---

## [Editor Report · Decision Letter 1]

12 Oct 2022

The Relationship between Trust and Attitudes towards the COVID-19 Digital Contact-Tracing App in the UK

PONE-D-22-17800R1

Dear Dr. Dowthwaite,

We’re pleased to inform you that your manuscript has been judged scientifically suitable for publication and will be formally accepted for publication once it meets all outstanding technical requirements.

Kind regards,

Mukhtiar Baig, Ph.D.

Academic Editor

PLOS ONE

---

## [Editor Report · Acceptance letter]

19 Oct 2022

PONE-D-22-17800R1 

The Relationship between Trust and Attitudes towards the COVID-19 Digital Contact-Tracing App in the UK 

Dear Dr. Dowthwaite:

I'm pleased to inform you that your manuscript has been deemed suitable for publication in PLOS ONE. Congratulations! Your manuscript is now with our production department. 

Kind regards, 

on behalf of

Professor Mukhtiar Baig 

Academic Editor

PLOS ONE